# Symptomatic and Asymptomatic Protist Infections in Hospital Inpatients in Southwestern China

**DOI:** 10.3390/pathogens10060684

**Published:** 2021-05-31

**Authors:** Shun-Xian Zhang, David Carmena, Cristina Ballesteros, Chun-Li Yang, Jia-Xu Chen, Yan-Hong Chu, Ying-Fang Yu, Xiu-Ping Wu, Li-Guang Tian, Emmanuel Serrano

**Affiliations:** 1Clinical Research Center, LongHua Hospital Shanghai University of Traditional Chinese Medicine, Shanghai 200032, China; zhangshunxian110@163.com; 2Chinese Center for Tropical Diseases Research, National Institute of Parasitic Diseases, Chinese Center for Disease Control and Prevention, Shanghai 200025, China; chenjx@nipd.chinacdc.cn (J.-X.C.); chuyh@nipd.chinacdc.cn (Y.-H.C.); yuyf@nipd.chinacdc.cn (Y.-F.Y.); wuxp@nipd.chinacdc.cn (X.-P.W.); 3Parasitology Reference and Research Laboratory, National Centre for Microbiology, Health Institute Carlos III, Majadahonda, 28220 Madrid, Spain; dacarmena@isciii.es; 4Wildlife Ecology & Health Group (WE&H), Servei d’Ecopatologia de Fauna Salvatge (SEFaS), Departament de Medicina i Cirurgia Animals, Universitat Autònoma de Barcelona (UAB), E-08193 Bellaterra, Spain; cballesteros6@rvc.ac.uk; 5Department of Clinical Research, the 903rd Hospital of People’s Liberation Army of China, Hangzhou 310013, China; chunliyang2014@163.com; 6NHC Key Laboratory of Parasite and Vector Biology, National Institute of Parasitic Diseases, Chinese Center for Disease Control and Prevention, Shanghai 200025, China; 7School of Global Health, Chinese Center for Tropical Diseases Research-Shanghai Jiao Tong University School of Medicine, Shanghai 200025, China

**Keywords:** coinfection, enteric protists, China, *Giardia duodenalis*, *Entamoeba histolytica*, *Cryptosporidium*, *Blastocystis* sp., genotyping, molecular diversity

## Abstract

*Cryptosporidium* spp., *Entamoeba histolytica*, *Giardia duodenalis*, and *Blastocystis* sp. infections have been frequently reported as etiological agents for gastroenteritis, but also as common gut inhabitants in apparently healthy individuals. Between July 2016 and March 2017, stool samples (*n* = 507) were collected from randomly selected individuals (male/female ratio: 1.1, age range: 38–63 years) from two sentinel hospitals in Tengchong City Yunnan Province, China. Molecular (PCR and Sanger sequencing) methods were used to detect and genotype the investigated protist species. Carriage/infection rates were: *Blastocystis* sp. 9.5% (95% CI: 7.1–12.4%), *G. duodenalis* 2.2% (95% CI: 1.1–3.8%); and *E. histolytica* 2.0% (95% CI: 0.9–3.6%). *Cryptosporidium* spp. was not detected at all. Overall, 12.4% (95% CI: 9.7–15.6) of the participants harbored at least one enteric protist species. The most common coinfection was *E. histolytica* and *Blastocystis* sp. (1.0%; 95% CI: 0.3–2.2). Sequence analyses revealed that 90.9% (10/11) of the genotyped *G. duodenalis* isolates corresponded to the sub-assemblage AI. The remaining sequence (9.1%, 1/11) was identified as sub-assemblage BIV. Five different *Blastocystis* subtypes, including ST3 (43.7%, 21/48), ST1 (27.1%, 13/48), ST7 (18.8%, 9/48), ST4 (8.3%, 4/48), and ST2 (2.1%, 1/48) were identified. Statistical analyses confirmed that (i) the co-occurrence of protist infections was purely random, (ii) no associations were observed among the four protist species found, and (iii) neither their presence, individually or jointly, nor the patient’s age was predictors for developing clinical symptoms associated with these infections. Overall, these protist mono- or coinfections are asymptomatic and do not follow any pattern.

## 1. Introduction

Parasitic infections have been frequently reported as significant causes of gastrointestinal disorders and major contributors to the global burden of diarrheal disease globally [1,2,3,4,5]. Several diarrhea-causing enteric parasite species have been described in humans. Among them, the most relevant are the protozoa *Cryptosporidium* spp., *Entamoeba histolytica*, and *Giardia duodenalis*, and the stramenopile *Blastocystis* sp. [5,6]. However, the impact that these agents exert on public health has not been fully characterized yet as a large amount of the epidemiological information currently available is mainly based on single protist infections, and the role of coinfections is often understated. It is estimated that around 20% of child diarrheal episodes are reported in low and middle-income countries, and 9% of those documented in high-income settings are caused by *Cryptosporidium* spp. [1]. Amebiasis, the acute disease caused by *E. histolytica*, affects near 50 million people and causes 100,000 deaths each year [7]. Both *Cryptosporidium* spp. and *E. histolytica* have been recognized as significant causes of morbidity and mortality associated with diarrhea by the 2017 Global Burden Disease Study [4]. On the other hand, *G. duodenalis* infection is estimated to result in 280 million cases annually worldwide [8] and was included, together with *Cryptosporidium* spp., in the “Neglected Disease Initiative” launched by the World Health Organization in 2004 [9]. *Blastocystis* sp. is regarded as the most prevalent enteric protist isolated from diarrheal patients in high-income countries [1]. However, its pathogenicity remains controversial partly because it is also the most frequent non-fungal eukaryotic organism detected in fecal samples from apparently healthy individuals [10,11,12]. It has been argued that this protist could be used as an indicator of potential exposure to other pathogenic enteric protozoa [13]. Besides *Blastocystis* sp., *Cryptosporidium* spp., *E. histolytica*, and *G. duodenalis* infections are common in apparently healthy individuals [1,14,15]. There is a lack of reliable data on the true burden of asymptomatic infections due to the absence of monitoring programs, underreporting and the fact that carriage of subclinical stages is often underdiagnosed [1,16,17].

Even though there have been notable advances in this field, the factors that determine the course of enteric protozoan infections and the development (or not) of gastrointestinal symptoms remain poorly understood [18]. Protist species/genotypes have been suggested as predictors of pathogenicity/virulence. It is expected that the growing development of new molecular diagnostic tools contributes to clarifying the distinction between pathogenic and nonpathogenic lineages, as well as pathophysiological interactions and other epidemiological features of interest [19].

Many previous studies have explored and reported the presence of more than one pathogen in cases with diarrheal disease [6,19,20] as well as in healthy individuals [21,22]. Comparatively, less effort has been devoted to exploring the impact of concomitant enteric infections involving viral, bacterial, and/or parasitic agents and characterizing the enteric communities and the existence of specific interactions among them [15,23,24]. These interactions, at least between 2 specific enteric pathogens, are well documented in the veterinary field, such as the association of enterotoxigenic *Escherichia coli* and rotavirus that lead to severe diarrhea in piglets as an example [25]. However, it has not been addressed in immunocompetent humans until recently [15,26].

To improve our current understanding of the existence and impact of intestinal protozoan coinfections in humans, we conducted a hospital-based cross-sectional study. The aims of the survey were: (i) to investigate the occurrence of the four protist species most commonly associated with gastrointestinal disorders and to determine their molecular profile, (ii) to assess whether these protists co-occur by chance or are community-structured in hospital-based patients, and (iii) to explore the potential impact of coinfections on the clinical symptomatology among immunocompetent patients attending the People’s Hospital of Tengchong City and the Chinese Medicine Hospital in Tengchong City.

## 2. Results

### 2.1. Characteristics of the Study Population

A total of 507 subjects participated in the study. The male:female ratio was 1.1 (260/247). Han nationality was predominant (94.9%, 481/507), and the median age of recruited participants was 52 years (interquartile range (IQR): 38–63 years). Most of the subjects lived in rural areas (78.1%, 396/507). Around two-thirds of them had completed primary education (65.6%, 332/507), and 63.1% of the participants were farmers (320/507).

The most common clinical symptom was decreased appetite (17.9%, 91/507), followed by abdominal pain (16.6%, 84/507) and nausea (16.6%, 84/507); in addition, 10.4% (59/507) of recruited patients presenting with acute diarrhea. Other less frequently reported symptoms included abdominal distension (12.4%, 63/507), itchy skin 12.2%, 62/507), constipation (10.8%, 55/507) and perianal pruritus (5.5%, 28/507). Overall, 50.0% (233/507) of the patients did not have any gastrointestinal symptoms.

### 2.2. Single Enteric Pathogen Infections and Coinfections

*Blastocystis* sp. was the most prevalent protist species found (9.5%, 48/507; 95% CI: 7.1–12.4), followed by *G. duodenalis* (2.2%, 11/507; 95% CI: 1.1–3.8) and *E. histolytica* (2.0%, 10/507; 95% CI: 0.9–3.6), whereas *Cryptosporidium* spp. was not detected at all. *Entamoeba histolytica* was more frequently found in diarrheal (11.9%, 7/59) than in non-diarrheal (0.7%, 3/448) cases (OR = 19.9, 95% CI: 5.0–79.5). The same was true for *G. duodenalis* (6.8%, 4/59 versus 1.6%, 7/448; OR = 4.6, 95% CI: 1.3–16.2). In contrast, no significant differences were observed on the distribution of *Blastocystis* sp. between diarrheal and non-diarrheal cases (13.6%, 8/59 versus 8.9%, 40/448; OR = 1.6, 95% CI: 0.7–3.6).

Overall, 12.4% of patients (63/507, 95% CI: 9.7–15.6) were infected by at least one enteric protist. Infections caused by two different enteric protist species were detected in 1.2% of patients (6/507, 95% CI: 0.4–2.6). The most common coinfection found was *E. histolytica* and *Blastocystis* sp. (1.0%, 5/507; 95% CI: 0.3–2.2), and the coinfection of *E. histolytica* and *Blastocystis* sp. in diarrheal cases was more common than in non-diarrheal patients (5.1%, 3/59; 0.4%, 2/448, chi-squared = 10.8, *p*-value = 0.001). Coinfection by *G. duodenalis* and *Blastocystis* sp. was identified in a single case (0.2%, 1/59; 95% CI: 0.01–1.1), and no significant difference of the coinfection by *G. duodenalis* and *Blastocystis* sp. was found in subjects with and without diarrhea (0.2%, 1/59; 0.0%, 0/449, chi-squared = 2.81, *p*-value = 0.116). No other coinfection of these four enteric protozoa was found in diarrhea individuals and healthy controls.

### 2.3. Genetic Characterization of Isolates

A total of 11 *G. duodenalis* isolates were successfully characterized at the triosephosphate isomerase (*tpi*) locus (Table 1). Sequence analysis allowed identifying assemblages A (90.9%, 10/11) and B (9.1%, 1/11). Further, all 10 assemblage A isolates were assigned to the sub-assemblage AI of the parasite, and the only one assemblage B isolate was identified as sub-assemblage BIV. *Giardia duodenalis* assemblage A was significantly more prevalent in individuals presenting with diarrhea (4/59) than in individuals without clinical manifestations (6/448, chi-squared = 7.37, *p*-value = 0.006; OR = 5.4, 95% CI: 1.5–19.6).

*Blastocystis* sp. was detected by PCR amplification of the small subunit ribosomal RNA (*ssu* rRNA) gene in 48 samples (Table 2). Sequence analyses revealed the presence of five subtypes (ST) of this protist, with ST3 being the most prevalently found (43.7%, 21/48), followed by ST1 (27.1%, 13/48), ST7 (18.8%, 9/48), ST4 (8.3%, 4/48), and ST2 (2.1%, 1/48). Concerning intra-subtype genetic diversity, alleles 2, 4, and 88 were identified within ST1, allele 9 within ST2, allele 34 within ST3, alleles 42, 92, and 94 within ST4, and alleles 100 and 101 within ST7. No statistical differences in the distribution frequencies of *Blastocystis* STs were observed between diarrheal and non-diarrheal individuals (chi-squared = 3.73, *p*-value = 0.2913).

### 2.4. Impact of Coinfection on Diarrheal Symptomatology

According to our partial least square (PLS) analyses, no statistically significant associations were identified between the presence of *G. duodenalis*, *Blastocystis* sp. and *E. histolytica*, alone or in combination, and the occurrence of clinical manifestations (Stone-Geisser’s Q2 test value <0.0975). Concerning the other predictor, age was not statistically associated with symptomatology in the study population either.

Furthermore, the presence of any of the three protist species investigated, jointly or separately, covaried negatively with symptomatology. The analysis revealed that most of the X’s component variance was due to coinfection with enteric protists (43.1%), followed by *Blastocystis* sp. (27.6%) and *E. histolytica* (24.0%) (Table 3).

### 2.5. Co-Occurrence of Enteric Protozoa

The null model analysis showed that the observed C-score (265) was lower than expected by chance (267.45), indicating the existence of a random, noncompetitively structured protist community. No evidence of a statistically significant protist combination could be demonstrated in the surveyed population (SES = −0.33, *p*-value = 0.414).

## 3. Discussion

Our study confirmed the single prevalence rates of enteric protist species, including *G. duodenalis*, *Blastocystis* sp., and *E. histolytica* commonly reported in previous studies in similar settings characterized by adequate access to water sanitation and personal hygiene practices (WASH) [1,17,27]. In the present PCR-based report, *Blastocystis* sp. was the predominant intestinal protist species identified, followed by *G. duodenalis* and *E. histolytica*, whereas *Cryptosporidium* spp. was not detected at all. These data were also consistent with those documented in previous studies conducted in human populations in China [28].

Regarding the genetic diversity of the protist species investigated, out of the 11 *G. duodenalis* isolates successfully genotyped at the *tpi* locus, 10 (90.9%) were classified as assemblage A and the remaining one (9.1%) as assemblage B. All assemblage A sequences were identified as sub-assemblage AI. This finding is interesting as sub-assemblage AI is typically reported at much lower frequencies than sub-assemblage AII in European [29,30,31] and Asian [32] countries. This discrepancy may be due to differences in infection sources or transmission pathways (e.g., a variable proportion of infections of zoonotic nature). No mixed coinfections involving assemblages A and B were found. Similarly, no infections caused by animal-specific assemblages C–D (dogs), E (domestic and wild ungulates) and F (cats) were detected, suggesting that these host species play a limited role as a source of human giardiasis in this geographical area.

It has been suggested that the presence and proportion of *G. duodenalis* assemblages may present spatiotemporal variations [33,34]. Additionally, socioeconomic factors have also been suggested as potential drivers for *G. duodenalis* assemblage distribution [35]. Interestingly, some molecular epidemiological investigations have reported that assemblage’s segregation may be involved in infection outcome and clinical presentation, with assemblage B resulting in more frequently symptomatic infection in endemic settings [21,36]. Other surveys have indicated that sub-assemblage AII may represent a more virulent genetic variant of *G. duodenalis* in humans [17,37,38]. The data mentioned above may explain, at least partially, the lack of evidence in support of a potential association between the presence of *G. duodenalis* and the occurrence of clinical symptomatology in the present study. It should be noted, however, that a recent case–control study assessing the frequency and genetic diversity of *G. duodenalis* infections in children younger than five years of age with and without diarrhea in southern Mozambique has demonstrated that the occurrence of gastrointestinal illness was not associated with a given genotype of the parasite [39].

Five distinct *Blastocystis* STs were identified in the hospital inpatient population under study, including ST1–ST4 and ST7. This is well in agreement with the available molecular data from China, where ST1–ST7 and ST12 have been described at variable frequency rates in different human populations [40]. Interestingly, clinical studies on patients presenting with diarrhea have shown that ST1 was related to clinical manifestations, including diarrhea and may have, therefore, potential pathogenicity. In contrast, ST3 was the *Blastocystis* genetic variant more frequently identified in asymptomatic infections [41,42,43]. This potential link between a given *Blastocystis* ST and the presence/absence of clinical manifestations could not be demonstrated in the present survey. Of note, a relatively high proportion of individuals carried *Blastocystis* ST7, a subtype mostly identified in birds and rarely found in mammals, including humans [44]. This finding clearly indicates that contact with poultry or captive avian species (or with their fecal material) was the most likely source of infection of these inpatients.

Remarkably, *Cryptosporidium* spp. was not detected in any of the recruited patients in this survey. In China, an overall *Cryptosporidium* infection rate of 3% has been estimated for 1987‒2018 [45]. In hospital settings, documented infection rates vary greatly depending on the clinical population under study. Cryptosporidiosis cases have been reported with low frequencies in children hospitalized primarily for non-gastrointestinal illnesses (1.6%, 102/6284) in Shanghai [46] and in diarrheic children (10/500) in Wuhan, Hubei province [47]. In contrast, comparatively much higher (6‒40%) infection rates have been identified in gastrointestinal (including esophageal, small intestine, colorectal, and liver) cancer patients [48]. These discrepancies may be associated with differences in the age and immunological status of the patients, the diagnostic methods used or even the geographical area considered.

As we have shown, it seems that, at the enteric protist level, there is not any specific community assemblage in the study population considering the four protist species considered. Our work showed that the occurrence of those pathogens was purely random and asymptomatic in the study population. Furthermore, not specific interactions were detected between them, and there was no evidence suggesting that their presence, jointly or separately, was associated with developing clinical symptomatology classically associated with infections by these species. Although all of them have been reported as causative agents of gastrointestinal disorders, especially in children in LMIC [20], asymptomatic carriage has also been commonly reported elsewhere. In the case of *Blastocystis* sp., its pathogenic relevance is still controversial [10,11] since this species is one of the most common enteric protists detected in humans [12,49]. Therefore, it is not surprising that their presence was not associated with clinical symptomatology in immunocompetent individuals.

Given the results obtained, different factors might provide an explanation that helps to understand the pathogenesis of enteric protozoan infections and what are the conditions that lead to disease. Some of them are related to the characteristics of the pathogen, such as different pathogenicity due to virulence variability of strains or the need of, at least, a second infection with another pathogen to cause clinical symptomatology. Moreover, host factors can play a key role, too, such as individual susceptibility or the presence of healthy functional barriers that protect the human intestine: the mucus layer, the intestinal epithelial layer and the intestinal microbiota [5,15,18,26]. However, further research needs to be performed to establish the ecology of enteric communities in healthy and unhealthy individuals.

We are aware that our research may have some limitations. First, this was a hospital-based cross-sectional study, so the surveyed population may not represent the general population in Tengchong City, and the generalizability to other populations may be rather limited. Although the occurrence rates of protist enteroparasites were consistent with those published in previous studies, low prevalences might have negatively influenced the reliability of the estimates in the PLS analyses. Second, the molecular methods (direct and nested PCRs) used here did not allow the quantification of parasites’ DNA, so the direct association between parasites’ burdens and occurrence of clinical manifestations could not be determined. In addition, acute bacterial or viral infections of the gastrointestinal tract were not found in this study. Additionally, molecular data on the diversity and frequency of species/genotypes were based on single-locus sequence analysis. Adopting multi-locus genotyping schemes (particularly for *G. duodenalis*) would likely improve the genetic data provided here. This is a task to be conducted soon.

Therefore, further research with a larger sample size, which considers a broader enteropathogen community, is needed to explore community assemblages present and their relationships with pathogenicity and clinical manifestations in rural and urban regions in China.

Undoubtedly, with the increasing awareness of the microbiota role in developing host immunity [5] and their potential relationship with many communicable and noncommunicable diseases, such as irritable bowel syndrome [50], there is a growing need to understand the complex relationships between different pathogens and as well as between the intestinal microbiota and pathogens.

## 4. Materials and Methods

### 4.1. Study Design and Study Area

A cross-sectional hospital-based study was conducted from July 2016 to March 2017 in Tengchong (25°01′15″ N, 98°29′50″ E, 1596 m above sea level), a county-level city located in Yunnan Province, Southwest China. Tengchong has a tropical monsoon climate. The annual average temperature is 15 °C, and the average annual rainfall is 1535 mm with a year-round mean relative humidity of 77%. The total resident population is 659,000 (Census 2014), of which 60.5% live in rural areas. Two hospitals from this city (People’s Hospital of Tengchong City and Chinese Medicine Hospital of Tengchong City) agreed to participate in the study.

The target sample size was *n* = 423 for an expected prevalence of 50%, 95% confidence interval and 5% precision. Finally, 507 patients were recruited.

### 4.2. Study Participants

Voluntary hospital inpatients (diarrhea cases and healthy controls) were recruited after a clear explanation of the study objectives provided by a member of the researcher team through personal interviews. Informed consent was obtained from the participants or their parents/legal guardians. Individuals presenting with diarrhea were recruited from the Gastroenterology Department. Non-diarrheal individuals were recruited from other hospital departments, including Respiratory Department, Trauma Department, and Emergency Department, among others. Enrolled patients were immunocompetent individuals with CD4 lymphocyte counts ≥500 cells/mm^3^.

According to the World Health Organization, an acute diarrhea case was defined as an individual who had more than three episodes of abnormal stool within 24 h (e.g., loose, watery, bloody, or mucous stool) for any period lasting for <14 days [15]. A healthy subject was defined as a person who did not have any diarrhea symptoms in the 14 days before recruitment into the study. Subjects who met the following inclusion criteria were eligible for inclusion in this study: (i) patients presenting with acute diarrhea (cases), (ii) patients without gastrointestinal manifestations, including diarrhea (controls), (iii) patients with a negative result to the main bacterial (*Campylobacter* spp., *Clostridium* spp., *Escherichia coli*, *Salmonella* spp., *Shigella* spp., and *Yersinia enterocolitica*) and viral (rotavirus, adenovirus, sapovirus, norovirus, and astrovirus) enteric pathogens, and (iv) patients, from which written informed consents were available. Exclusion criteria included (i) subjects providing insufficient amount of fecal material, (ii) subjects having severe disorders of the cardiovascular or nervous systems, serious mental diseases, tumors, hepatitis A–C, and E virus infection, and/or human immunodeficiency virus (HIV) infection/acquired immune deficiency syndrome (AIDS), (iii) subjects, who did not provide written informed consent, (iv) lactating women, and (v) subjects taking antiparasitic drugs just before or during the recruitment period.

### 4.3. Specimen and Data Collection

Single stool samples (>3 g or >3 mL) were obtained from each participating subject using a sterile sampling cup during the study period at the two hospital settings and stored at 4 °C. Collected samples were shipped to the laboratory of the National Institute of Parasitic Diseases, Chinese Center for Disease Control and Prevention (Shanghai, China) without interrupting the cold chain and immediately stored at −70 °C until further processing.

A structured questionnaire was used to gather sociodemographic (including age, gender, educational level, occupation, nationality, and place of residence) and clinical (clinical manifestations associated with enteric protist infections, including abdominal distension, diarrhea, lack of appetite, itchy skin, perianal pruritus, constipation, nausea, abdominal pain, number of stools per day and type of stools) data from each recruited patient.

### 4.4. DNA Extraction and Purification

Genomic DNA was extracted from each stool sample (0.2 g or 0.2 mL) using a QIAamp DNA stool mini kit (Qiagen, Hilden, Germany) according to the manufacturer’s protocol. Purified genomic DNA was stored at −70 °C until downstream analysis by polymerase chain reaction (PCR) amplification.

### 4.5. Molecular Detection of Intestinal Protozoan Species

A direct PCR protocol was used to detect *Blastocystis* sp. [51]. In contrast, nested PCR protocols were used for identifying *Cryptosporidium* spp., *G. duodenalis*, and *E. histolytica* [52,53,54]. The description of the primer pairs used, the expected size of the obtained amplicons and the cycling conditions of these protocols are provided in Appendix A.

### 4.6. Sanger Sequencing Analysis

Positive-PCR products of the expected sizes were directly sequenced in both directions using appropriate internal primer sets (Appendix A). DNA sequencing was conducted at Sangon Biotech Company (Shanghai, China). Raw sequencing data were viewed using the Chromas Lite version 2.1 sequence analysis program (https://technelysium.com.au/wp/chromas/. Accessed on 19 May 2021). The BLAST tool (http://blast.ncbi.nlm.nih.gov/Blast.cgi. Accessed on 19 May 2021) was used to compare nucleotide sequences with sequences retrieved from the NCBI GenBank database. Generated DNA consensus sequences were aligned to appropriate reference sequences using the MEGA 6 software to identify *G. duodenalis* species and assemblages/sub-assemblages. *Blastocystis* sequences were submitted to the *Blastocystis* 18S database (http://pubmlst.org/blastocystis/. Accessed on 19 May 2021) for sub-type confirmation and allele identification. The sequences obtained in this study were deposited in GenBank under accession numbers MW810321‒MW810323 (*G. duodenalis*) and MW798733‒MW798742 (*Blastocystis* sp.).

### 4.7. Statistical Modeling

Data were analyzed using the software R version 4.0.5 [55]. The chi-squared or Fisher’s exact test, Odd ratios (OR) and 95% confidence intervals (95% CIs) were used to compare and describe the qualitative variables.

Only participants with complete data records were included in the final analysis. A new variable was created according to the World Health Organization’s definition of diarrhea and the information gathered about the type of stool and the number of depositions per day. A case of acute diarrhea was defined as a person with more than three episodes of liquid stools per day, lasting less than 2 weeks [56]. Prevalence rates at 95% CIs for single infections and coinfections in the study population were calculated using epiR library version 0.5‒10 [57].

#### 4.7.1. Co-Occurrence of Enteric Pathogens

Null model analysis was used to explore whether enteric protozoa coinfections were positive, negative, or randomly associated. Data were organized as a presence-absence 4 × 507 (row × columns) matrix, in which each row represented a protozoa species. Each column represented a study participant, “1” indicated that a species was present at a particular host and “0” indicated that a species was absent.

The C-score was the co-occurrence index used for co-occurrence patterns characterization. The algorithm chosen was the fixed row-equiprobable column [58]. The calculated C-score was compared with the expected C-score calculated for 5000 randomly assembled null matrices by Monte Carlo simulations. Furthermore, to compare the degree of co-occurrence across data, a standardized effect size (SES) was calculated, an index that measures the number of standard deviations that the observed index (C-score) is above or below the mean index of the simulated communities. The package “EcoSimR” version 0.1.0 was used to carry out the analysis [59].

#### 4.7.2. Assessing the Impact of Coinfection with Enteric Pathogens on Diarrhea Severity

The partial least square (PLS) regression method was used to assess the impact of coinfection with enteric protozoa on developing clinical symptomatology. This technique was selected as it offers multiple advantages over other regression methods: it is the least restrictive of the multivariate techniques for exploring complex ecological patterns [60], including the impact of coinfections on the host’s health [23], and its distribution is free and well suited to deal with multicollinearity [61]. In our analysis, we defined explanatory and response components or blocks. The explanatory block (PLS X’s component) was defined by a presence-absence matrix representing the enteric protozoa community (*Blastocystis* sp., *G. duodenalis*, *E. histolytica*, and *Cryptosporidium* spp.). In addition, due to the previously mentioned age variability in the clinical presentation of diarrheal diseases, age in years was also included as a covariate in the explanatory block. Our response block (PLS’s Y component) included the main symptoms described associated with the infections of those protist species (abdominal distension, lack of appetite, itchy skin, perianal pruritus, constipation, nausea, abdominal pain and acute diarrhea).

The significance of PLS models was assessed using Stone–Geisser’s Q2 test, a cross-validation redundancy measure created to evaluate the predictive significance of exogenous variables. Values greater than 0.0975 indicate that predictors are statistically significant, whereas values below this threshold reveal no significance. Finally, the percentage of observed MNLS variability explained by the enteric pathogen block was also estimated. The “plspm” version 0.4.9 was used to perform the analysis [62].

## 5. Conclusions

The present study was the first to analyze the community assemblage of the four protist species commonly associated with human gastrointestinal disorders in immunocompetent individuals. Our results showed the absence of any structured community between them. Their occurrence was purely random. Moreover, there was no evidence of an association between their presence and developing clinical symptomatology.

Further research, including a broad range of enteric pathogens, is needed to disentangle the complex relationships and interactions of the intestinal ecosystem. This could eventually lead to a better understanding about what are the drivers behind gastrointestinal disorders.

## Figures and Tables

**Table 1 pathogens-10-00684-t001:** Diversity, frequency, and molecular features of *Giardia duodenalis* sequences at the *tpi* locus generated in the present study. GenBank accession numbers are provided.

Locus	Assemblage	Sub-Assemblage	Isolates	Reference Sequence	Stretch	Single-Nucleotide Polymorphisms	GenBank ID
*tpi*	A	AI	1	L02120	594–1036	None	MW810321
			1	L02120	586–1060	G746A, A834G	MW810322
			1	L02120	595–1060	G915A, G996A, G1004	MW810323
			7 ^1^	L02120	‒	‒	‒
	B	BIV	1 ^1^	AF069560	‒	‒	‒

^1^ isolates with associated sequences of insufficient quality to clearly determine the presence of single nucleotide polymorphisms.

**Table 2 pathogens-10-00684-t002:** Diversity, frequency, and molecular features of *Blastocystis* sp. sequences at the *ssu* rRNA locus generated in the present study. GenBank accession numbers are provided.

Subtype	Allele	Isolates	GenBank ID
ST1	2	3	MW798733
	4	7	MW798734
	88	3	MW798735
ST2	9	1	MW798736
ST3	34	21	MW798737
ST4	42	2	MW798738
	92	1	MW798739
	94	1	MW798740
ST7	100	3	MW798741
	101	6	MW798742

**Table 3 pathogens-10-00684-t003:** Predictor weights of the PLS model explaining their association with clinical symptomatology in hospital-based patients in Tengchong City.

Predictor Variables	Loads	Weights	Percent	Cross-Correlation
Protozoa richness	−0.67	−0.66	43.1	−0.11
*Giardia duodenalis*	−0.2	−0.17	2.9	−0.03
*Blastocystis* sp.	−0.56	−0.53	27.6	−0.08
*Entamoeba histolytica*	−0.44	−0.5	24	−0.08

## Data Availability

All relevant data are within the article and its additional files. The sequences obtained in this study were deposited in GenBank under accession numbers MW810321-MW810323 (*Giardia duodenalis* at the *tpi* locus) and MW798733‒MW798742 (*Blastocystis* sp.).

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
