# Peer review of "Symptomatic and Asymptomatic Protist Infections in Hospital Inpatients in Southwestern China"

_pathogens, 2021, doi:10.3390/pathogens10060684_

Round 1

Reviewer 1 Report

The article entitled "Symptomatic and asymptomatic protist infections in hospital inpatients in southwestern China" by Tian and collaborators investigates the occurrence and co-occurrence of protists in the gastrointestinal tract of patients in sentinel hospitals in China, and the possible correlation between the presence of these pathogens and symptoms manifestations. The study is based on molecular approaches to  detect and genotype those pathogens. The statistical analysis did not show correlation between the presence of protists and presence of symptoms. 

My overall comments for the authors is for a more deep study involving host elements and protists pathogenicity factors. In this way, you will be able to answer the question about presence of protists and symptoms. I am positive about it. If you don't have the right structures to do that, I recommend reshape your question for a more epidemiological setting and present all the data possible (in tables or another creative form) in a way to launch some interesting questions or direction.Nonetheless, I recognized the afford of each author in this study. 

Reviewer 2 Report

The manuscript “Symptomatic and asymptomatic protist infections in hospital 2 inpatients in southwestern China” is a well written manuscript in an important issue for the journal namely the existence and impact of intestinal protozoan co-infections in humans. An hospital-based cross-sectional study was performed in two different Chinese hospitals, using molecular diagnosis techniques. I would suggest below changes in order to clarify the document.

Abstract:

Line 27- please refer the age range and sex ratio.

Line 35 : Author have identified five different Blastocytis types and not four.

A last sentence with conclusion is missing in the abstract.

Introduction is well written and the aims well defined.

Results In pint 2.1 I would suggest in line 108 to refer the number (percentage) of cases with diarrhea according to WHA definition.

Table 1 In locus tpi, assemblage A/AI second line there is an error in the stretch (is 5864, and should be 500 and not 5000)

Line 141 – Please clarify the sentence “between symptomatic and asymptomatic” Doers it means just diarrhea or other symptoms.

Discission : There is any reference to the fact that since the authors used molecular techniques there is no quantitative information (at least there is any reference to that). This should be discussed since the symptoms could be related to burden of infection.

Also there is no information concerning the fact that Cryptosporidium was negative in all samples. Could it be a technical problem (mutation on primers site??) or it could be a true negative. Please discuss that.

Material and methods

The Sentence “A new variable was created according to the World Health Organization’s definition of diarrhoea and the information gathered about type of stool and number of depositions per day. A case of acute diarrhoea was defined as a person with more than three episodes of  liquid stools per day, lasting less than 2 weeks [51].” In lines 312 to 316 should be in item 4.3 when you mention the questionnaire.

Reviewer 3 Report

The paper is interesting and brings new informations, although with limitations, about the occurrence, molecular aspects and clinical profile of protist infections. Especially molecular data which were obtained are valuable.

The authors planned

 i) to investigate the occurrence of the four protist species most commonly associated with gastrointestinal disorders, and to determine their  molecular profile –

ii) to assess whether these protists co-occur by chance or are community-structured in hospital-based patients,

as a kind of the descriptive research above mentioned two tasks have been realized in a group of hospitalized patients

and iii) to explore the potential impact of coinfections on the clinical symptomatology among immunocompetent patients – the last one needs more explanation.

I have only one major critical remark. In fact we do not know exactly what were all the inclusion and exclusions criteria for recruitment, mainly there is a problem with describing of the patients (53,1%) who did not present any gastrointestinal disorders. The problem is if they were diagnosed with any chronic disease of the gastrointestinal tract, at the moment of the recruitment to the study without acute symptoms e.g. due to remission.

Did the authors exclude patients with acute bacterial or viral infections of the gastrointestinal tract? Did recruited patients received any potent antiparasitic drugs before recruitment?

In my opinion authors should emphasize that their results refer to links between protist infections and acute disorders of gastrointestinal tract.

The study group was limited only to hospitalized patients. It is not known what was the reason of the admission to the hospital for those without symptoms from the digestive tract. How the state of immunocompetence was defined?

The authors informed that patients infected with HIV, HBV were excluded. I wonder what about chronic infection with HCV and acute disease caused by HAV, HEV and finally what about the impact of infections with hepatotropic viruses  on the development of gastrointestinal symptoms.

Round 2

Reviewer 1 Report

No comments.

Reviewer 2 Report

The referee acknowledges the authors effort in answer and perform or justify all the suggestions made.

I consider that the manuscript is now fine for publication.

Reviewer 3 Report

I accept authors' response.